# Research on Short-Term Load Forecasting Based on Optimized GRU Neural Network

Chao Li [ID], Quanjie Guo [ID], Lei Shao *, Ji Li and Han Wu

School of Electrical Engineering and Automation, Tianjin University of Technology, Tianjin 300384, China
* Correspondence: lei.shao@tjut.edu.cn

**Abstract:** Accurate short-term load forecasting can ensure the safe and stable operation of power grids, but the nonlinear load increases the complexity of forecasting. In order to solve the problem of modal aliasing in historical data, and fully explore the relationship between time series characteristics in load data, this paper proposes a gated cyclic network model (SSA–GRU) based on sparrow algorithm optimization. Firstly, the complementary sets and empirical mode decomposition (EMD) are used to decompose the original data to obtain the characteristic components. The SSA–GRU combined model is used to predict the characteristic components, and finally obtain the prediction results, and complete the short-term load forecasting. Taking the real data of a company as an example, this paper compares the combined model CEEMD–SSA–GRU with EMD–SSA–GRU, SSA–GRU, and GRU models. Experimental results show that this model has better prediction effect than other models.

**Keywords:** short-term load forecasting; set empirical mode decomposition; gated recurrent neural network; sparrow optimization algorithm

## 1. Introduction

With the rapid development of power systems, load forecasting attracted great attention of power companies and consumers, and became an important direction of modern power system research. Considering the periodicity, fluctuation, continuity, and randomness of power loads, the complexity and difficulty of load forecasting are increased.

Under the completely free power market operation mode, the load forecasting problem affects the power dispatching of power companies and the production plan of power-consuming enterprises [1]. Among them, short-term load forecasting plays an important role in guiding and regulating the operation of power companies. Accurate prediction results can more reasonably arrange the daily production plan. Short-term load forecasting (STLF) of the power system refers to forecasting the load in the next few hours to several days [2]. STLF is an important foundation to ensure the reliable operation of modern power systems and an important link in energy management systems. Its results play an important reference role for dispatching departments to determine the daily, weekly, and monthly dispatching plans, and to reasonably arrange the unit start–stop, load distribution, and equipment maintenance [3]. With the continuous expansion of the scale of modern power systems, higher requirements are put forward for STLF, and STLF technology of power systems is increasingly becoming a key technology in the power industry.

Short-term load forecasting methods can be divided into three categories: traditional forecasting technology, improved traditional technology, and artificial intelligence technology. Traditional techniques include regression analysis [4], least square method [5], and exponential smoothing method [6]. Improved technologies include time series method [7], autoregression and moving average based model [8], support vector machine [9], etc. However, most of the traditional technologies and improved traditional technologies are linear prediction models, and the relationship between load and other characteristic factors in load forecasting is complex and non-linear, so it is not effective in forecasting power load [10].

Artificial intelligence technology includes genetic algorithm [11], fuzzy logic [12], artificial neural network [13], and expert system [14]. However, these methods have some shortcomings. For example, artificial neural networks still need to extract features artificially in load forecasting, and human intervention is high.

In recent decades, with the rapid development of artificial intelligence technology, experts and scholars all over the world conducted in-depth research on short-term load forecasting and put forward many effective forecasting models. Imani, M [15] proposed a method to extract the non-linear relationship of load based on a convolutional neural network. Many researchers use deep learning networks to predict. Typical deep learning networks include CNN [16], deep confidence network, and recurrent neural network RNN [17]. CNN uses convolution operation to greatly reduce the data dimension and realize the learning and expression of data sample features. RNN is a special type of artificial neural network, which has a good ability to process sequence and time information. Kim, J et al. [18] combined RNN and CNN, and proposed a recursive starting convolutional neural network for load forecasting. When dealing with time series problems, the gradient of the recurrent neural network disappears, which makes it more difficult to train the RNN, resulting in poor prediction. Long- and short-term memory networks have been proposed to solve this problem [19]. However, the network structure of LSTM is complex and the convergence speed is slow. Gate-based loop architectures have been used to improve the LSTM, for example, a gated cycle unit (GRU). Compared to LSTM, GRU has one less gate function and requires fewer parameters, thus, improving the training speed. Wang, YX et al. [20] used GRU for short-term load prediction and achieved very good results.

The combination of different types of artificial neural network models is the research hotspot to solve the short-term power load forecasting problem. Rafi, SH et al. [21] proposed a neural network integrating CNN and LSTM for short-term load forecasting and achieved good results. However, the performance of CNN and LSTM should be further optimized. Shi, HF et al. [22] proposed a CNN–BiLSTM combination model optimized by attention mechanism for load forecasting. The combined model captures the data characteristics well and has a good prediction effect on long-time series. Gao, X et al. [23] proposed an EMD–GRU combined prediction model. The original sequence is decomposed by empirical mode decomposition and then predicted by GRU. EMD is prone to modal aliasing, and the combined model needs further improvement.

In addition, because it is very difficult to select super parameters in neural networks, it is easy to underfit or overfit, so parameter optimization is needed. In order to make the GRU model automatically find the optimal parameters in the training process, instead of human experience selection, the swarm intelligence optimization algorithm is used to optimize the parameter selection. Sparrow algorithm is one of the commonly used swarm intelligence optimization algorithms. Liao, GC [24] proposed an LSTM prediction model optimized by the sparrow algorithm. The prediction accuracy of LSTM model is effectively improved.

In summary, considering the signal noise problem in the original data sequence, this paper proposes to use the complementary set empirical mode decomposition to eliminate the noise interference in the original signal. Then the sparrow algorithm is used to optimize the parameters of GRU neural network. The CEEMD–SSA–GRU combination model is composed. Applying the model to short-term load forecasting, the network has higher forecasting accuracy and better adaptability.

## 2. Model Principle

### 2.1. GRU

In order to solve the problem that the feed forward neural network cannot retain previous information, some scholars proposed a feedback neural network, recurrent neural network (RNN), which can transfer the information between each layer in a two-way, and then form a memory for the information, allowing the information to persist, and has a certain memory capacity. Its structure is shown in Figure 1.

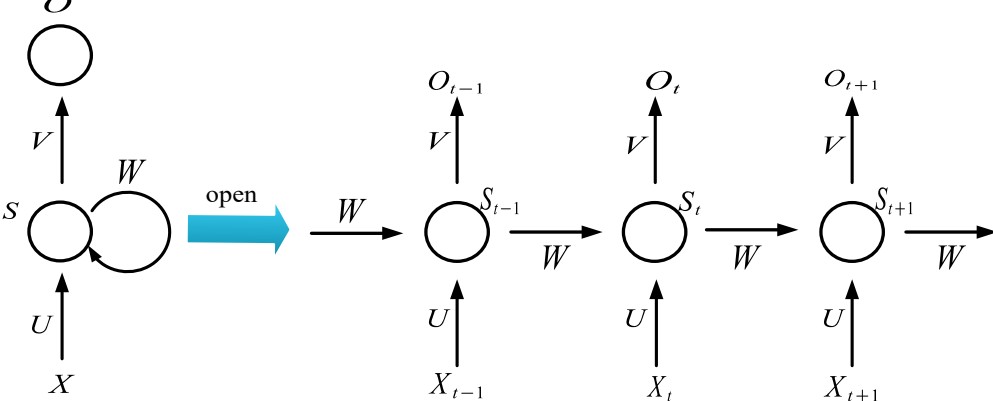

**Figure 1.** Structure diagram of RNN neural network.

As can be seen from Figure 1, $t$ represents the time, $X$ represents the input layer, $S$ represents the hidden layer, and $O$ represents the output layer. The calculation formula of $S$ and $O$ is as follows:

$$S_t = f(U \cdot X_t + W \cdot S_{t-1}) \tag{1}$$

$$O_t = g(V \cdot S_t) \tag{2}$$

Wherein $S_t$ in Formula (1) represents the hidden layer value at time $t$, $f(\cdot)$ represents the activation function of the hidden layer, $X_t$ represents the input vector at time $t$, $U$ represents the parameter matrix, $W$ represents the weight matrix, $S_{t-1}$ is the state of the hidden layer at the previous time. $O_t$ in Equation (2) represents the output at time $t$, $V$ represents the parameter matrix, and $g(\cdot)$ represents the activation function of the output layer. $g(\cdot)$ generally adopts softmax function, and $f(\cdot)$ can choose sigmoid function or tanh function.

Although RNN solves the problem that a feed forward neural network cannot remember information, RNN neural networks have some shortcomings. It can only deal with short-term dependence. It is difficult to solve long-term dependence when using RNN, and its memory capacity is limited. Moreover, when the sequence is long, its learning ability and memory ability decline, and there is the problem that the gradient disappears.

In order to solve the problems of RNN, a variant of RNN, long short-term memory neural network (LSTM), was proposed. LSTM not only solves the problems of RNN, but also can handle the problems of short-term and long-term dependence, and realizes the function of long-term and short-term memory. The network structure diagram of LSTM is shown in Figure 2.

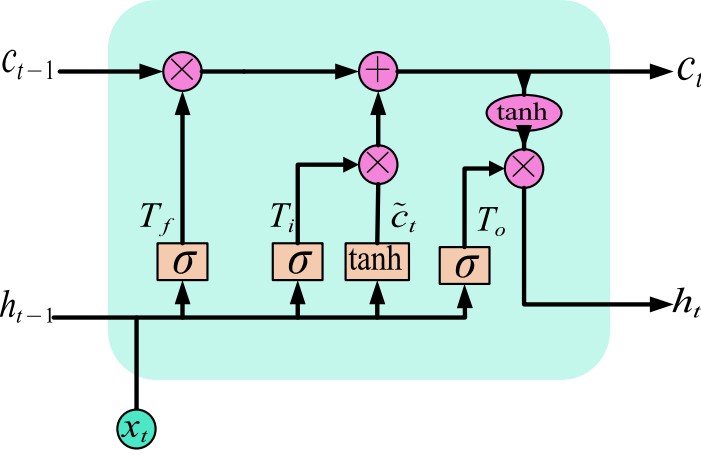

**Figure 2.** LSTM network structure.

It can be seen from Figure 2 that LSTM is much more complex in network structure than RNN, and LSTM introduces cell state $C_t$ to memorize information. At the same time, a gating structure is introduced to maintain and control information, i.e., input gate, forgetting gate, and output gate. Although LSTM solves the problem that RNNs cannot carry out long-term memory, the network structure of LSTM is complex and the convergence speed is slow. It affects the training process and results when carrying out power load forecasting, and causes problems such as training complexity. In order to solve these problems, a variant of LSTM, gated recurrent neural network (GRU), is proposed on the basis of LSTM. It optimizes the function of LSTM and makes the network structure simple. It is a widely used neural network at present. The structure of GRU has changed from the three gates of LSTM to two gates, i.e., update gate and reset gate. The network structure is shown in Figure 3.

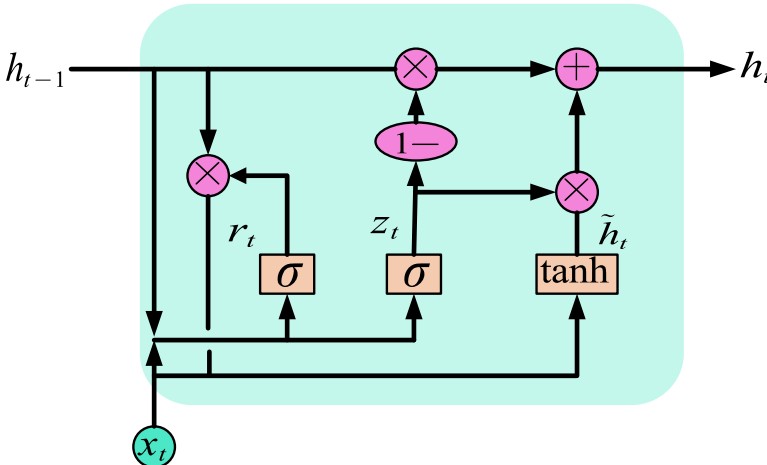

**Figure 3.** GRU network structure.

$z_t$ in Figure 3 means update door, $r_t$ represents the reset door. The function of these two gates is to control the degree to which information is transferred. The inputs of both gates are the input $x_t$ at the current time and the hidden state $h_t$ at the previous time. The calculation formula of the two doors is as follows:

$$z_t = \sigma(W_z \cdot [h_{t-1}, x_t]) \tag{3}$$

$$r_t = \sigma(W_r \cdot [h_{t-1}, x_t]) \tag{4}$$

where $x_t$ represents the input at time $t$, $h_{t-1}$ represents the hidden state at the previous time, [] represents the connection of two vectors, $W_z$ and $W_r$ represent the weight matrix, and $\sigma(\cdot)$ represents the sigmoid function.

GRU discards and memorizes the input information through two gate structures, and then calculates the candidate hidden state value $\widetilde{h}_t$, The calculation expression is shown in Formula (5):

$$\widetilde{h}_t = \tanh\left(W_{\widetilde{h}} \cdot [r_t \odot h_{t-1}, x_t]\right) \tag{5}$$

where $\tanh(\cdot)$ represents the tanh activation function, $W_{\widetilde{h}}$ represents the weight matrix, and $\odot$ represents the product of the matrix.

After the tanh activation function obtains the updated state information through the update gate, it creates vectors of all possible values according to the new input, and calculates the candidate hidden state value $\widetilde{h}_t$, Then, the final state $h_{t-1}$ at the current time is calculated through the network, as shown in Formula (6):

$$h_t = (1 - z_t) \odot h_{t-1} + z_t \odot \widetilde{h}_t \tag{6}$$

According to the above calculation formula, GRU stores and filters information through two gates, retains important features through gate functions, and captures dependencies through learning to obtain the best output value.

When the same effect is achieved, the training time of GRU is shorter. Especially in the case of large training data, the effect of training and prediction using GRU is better, and much time is saved. Therefore, this paper selects a GRU neural network model for short-term power load forecasting to achieve the purpose of short training time and good forecasting effect.

In the prediction process of the GRU model, the number of hidden layer neural units, the learning rate, the number of small batch training, and the number of iterations need to be considered. The values of these parameters can affect the model fitting effect, training duration, generalization ability, or degree of convergence. After many experiments and observing the loss value of the model, a set of empirical parameters are obtained. When the parameter selection of the GRU prediction model uses one hidden layer, the number of neurons is 50, the learning rate is 0.005, the data volume of batch training is 50, and the number of iterations is 100, and the GRU prediction model obtains relatively average calculation efficiency and prediction effect.

*2.2. Comparison of GRU Model and BP Model*

2.2.1. Error Evaluation Criteria

(1)    Mean absolute error (*MAE*)

$$MAE = \frac{1}{n} \sum_{i=1}^{n} |(\hat{y}_i - y_i)| \tag{7}$$

(2)    Root mean square error (*RMSE*)

$$RMSE = \sqrt{\frac{1}{n} \sum_{i=1}^{n} (\hat{y}_i - y_i)^2} \tag{8}$$

(3)    Mean absolute percentage error (*MAPE*)

$$MAPE = \frac{100\%}{n} \sum_{i=1}^{n} \left| \frac{\hat{y}_i - y_i}{y_i} \right| \tag{9}$$

2.2.2. Model Comparison

In this section, the GRU model and BP model are trained and predicted, and the prediction performance is compared. The power load data used in this paper are the real power load data of an industrial user's factory. The data set is selected from the real power load data of an industrial user in the two years from 1 January 2018 to 31 December 2020 as the training set data of the experiment. The sampling point is collected at the same time node every day, with a total of 731 sampling points. The daily power consumption load data in January 2021 are taken as the test set of the experiment, with a total of 31 data points. The environment of the electrical equipment in this factory is not affected by the external weather, and it is under constant temperature and humidity all the year round. The original power load data after missing or abnormal value processing are shown in Figure 4.

The normalized power load data is shown in Figure 5.

In this experiment, the GRU model and BP model are used to train the training set data. After the training, the data of the next month can be predicted. Finally, there is a comparison between the prediction results of the two models and the actual real value, using the above error evaluation indicators for analysis and comparison. The generated experimental results are shown in Table 1.

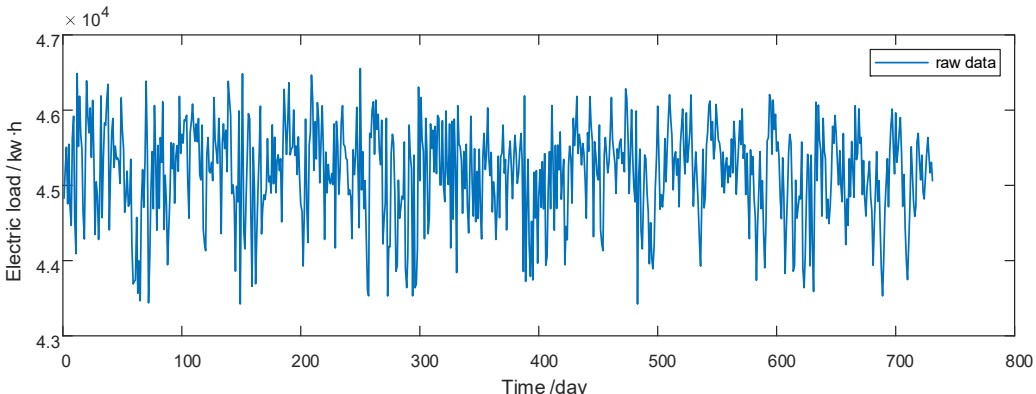

**Figure 4.** Original power load training data.

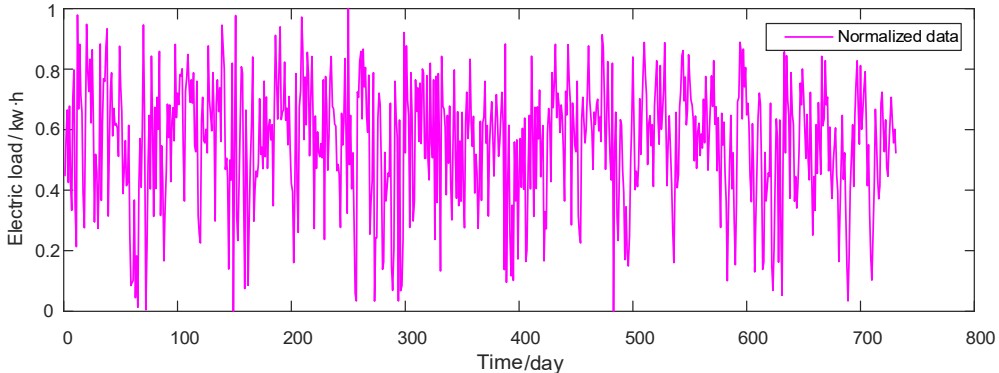

**Figure 5.** Normalized load data.

**Table 1.** Comparison of BP and GRU neural network prediction results.

| Time/ Days | Actual Value (kw·h) | BP Predicted Value (kw·h) | GRU Predicted Value (kw·h) | Time/ Days | Actual Value (kw·h) | BP Predicted Value (kw·h) | GRU Predicted Value (kw·h) |
|---|---|---|---|---|---|---|---|
| 1 | 45,360.37 | 49,005.88 | 44,884.87 | 17 | 45,424.44 | 44,644.86 | 45,362.05 |
| 2 | 46,338.27 | 45,771.37 | 45,536.35 | 18 | 46,514.46 | 45,562.02 | 45,358.09 |
| 3 | 45,877.85 | 45,362.05 | 45,899.23 | 19 | 45,238.45 | 46,166.70 | 45,106.73 |
| 4 | 45,942.52 | 47,015.26 | 45,789.56 | 20 | 45,856.65 | 45,677.46 | 45,829.11 |
| 5 | 45,840.91 | 45,021.95 | 45,886.01 | 21 | 45,862.76 | 45,046.72 | 45,227.82 |
| 6 | 46,291.17 | 44,214.40 | 45,753.38 | 22 | 45,593.45 | 46,841.13 | 44,584.44 |
| 7 | 45,471.49 | 46,052.58 | 45,018.84 | 23 | 45,831.85 | 44,945.20 | 44,296.32 |
| 8 | 45,142.91 | 44,868.51 | 44,456.50 | 24 | 45,343.99 | 46,129.58 | 44,143.25 |
| 9 | 44,379.54 | 46,287.86 | 44,890.29 | 25 | 44,872.57 | 45,047.65 | 44,303.67 |
| 10 | 45,039.70 | 45,801.54 | 46,133.09 | 26 | 46,133.70 | 45,341.62 | 45,317.62 |
| 11 | 44,262.69 | 45,572.41 | 45,215.21 | 27 | 44,743.78 | 45,592.95 | 43,627.78 |
| 12 | 45,438.92 | 44,993.34 | 45,650.29 | 28 | 45,927.41 | 44,866.74 | 44,493.00 |
| 13 | 45,054.37 | 45,178.43 | 45,961.51 | 29 | 45,654.40 | 44,784.20 | 44,050.53 |
| 14 | 44,940.83 | 45,213.52 | 45,521.12 | 30 | 45,462.24 | 45,267.86 | 44,076.97 |
| 15 | 45,883.15 | 44,737.06 | 46,267.10 | 31 | 45,460.24 | 45,833.42 | 44,504.96 |
| 16 | 44,214.98 | 45,417.60 | 45,714.39 | | | | |

It can be seen from Figure 6 that the trend of the GRU model curve is closer to the test curve and smoother. However, the BP model curve has a relatively turbulent trend, and it is difficult to judge local fluctuations in time and respond quickly. The main reason is that the BP neural network cannot remember and save information, while GRU has the function of long-term memory, which can better remember and store previous data. In order to more intuitively compare the prediction results of the two models, Figure 7 shows the numerical value and curve comparison of the prediction error of the model.

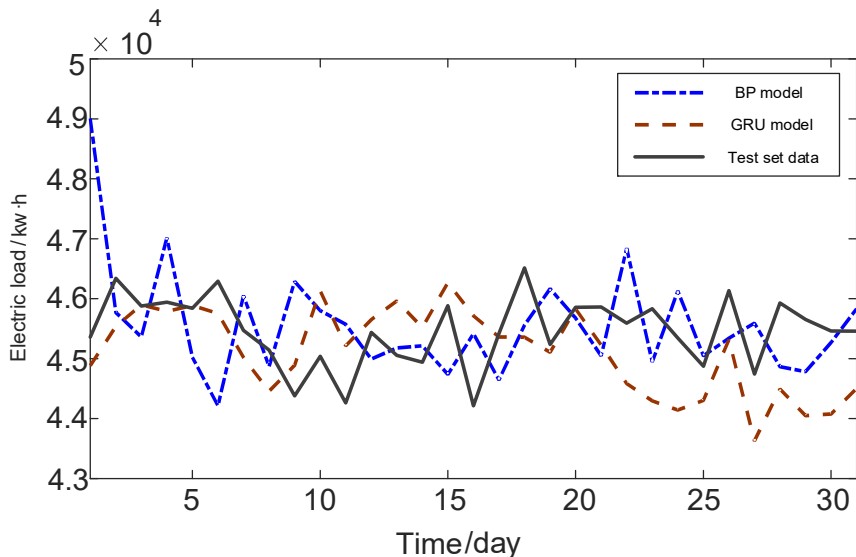

**Figure 6.** Comparison of prediction curves of BP and GRU models.

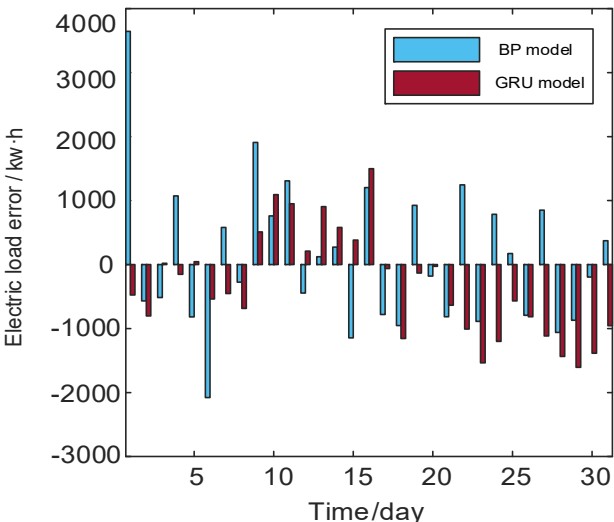

**Figure 7.** Error comparison between BP and GRU.

For the maximum relative error and the average absolute error, the BP model is 8.04% and 1.96%, respectively. The GRU model is 3.51% and 1.63%, respectively. Compared with BP model, the accuracy of the maximum relative error of GRU model is increased by 56.3%, and the accuracy of the average absolute error is increased by 16.8%. The GRU neural network has some problems in the training process. The GRU model has a slow speed in the training process, and its model parameters are obtained based on experience. It is easy to fall into local optimization, complicating the training process and increasing the difficulty of training.

### 2.3. Sparrow Search Algorithm

In order to make the GRU model automatically find the optimal parameters in the training process instead of manually selecting through experience, the intelligent optimization algorithm sparrow search algorithm (SSA) is used to optimize the parameters in the GRU model. A new swarm intelligence optimization algorithm sparrow search algorithm (SSA) was proposed, which is mainly affected by the sparrow's foraging and anti-predatory

behavior. Assuming that there are $n$ sparrows in a search space, the population composed of them can be expressed as:

$$X = \begin{bmatrix} x_{1,1} & x_{1,2} & \cdots & x_{1,m} \\ x_{2,1} & x_{2,2} & & x_{2,m} \\ & \vdots & \ddots & \vdots \\ x_{n,1} & x_{n,2} & \cdots & x_{n,m} \end{bmatrix} \tag{10}$$

where $m$ represents the dimension of the variable to be optimized. The fitness value of sparrow population can be expressed as:

$$F_x = \begin{bmatrix} f\left(\begin{bmatrix} x_{1,1} & x_{1,2} & \cdots & x_{1,m} \end{bmatrix}\right) \\ f\left(\begin{bmatrix} x_{2,1} & x_{2,2} & \cdots & x_{2,m} \end{bmatrix}\right) \\ \vdots \\ f\left(\begin{bmatrix} x_{n,1} & x_{n,2} & \cdots & x_{n,m} \end{bmatrix}\right) \end{bmatrix} \tag{11}$$

where $f$ represents the fitness value.

### 2.3.1. Update Discoverer Location

In the search process, discoverers with high fitness will obtain food first, and at the same time, they provide the followers with the area and direction where the food is located. Therefore, the search scope of the discoverer is wider and the search ability is stronger. The location update is described as follows:

$$X_{i,k}^{t+1} = \begin{cases} X_{i,k}^t \cdot \exp\left(-\frac{i}{a \cdot iter_{max}}\right), & R_2 < ST \\ X_{i,k}^t + Q \cdot L, & R_2 \geq ST \end{cases} \tag{12}$$

where $t$ is the current iteration number, $k = 1, 2, 3, \ldots, m$; $iter_{max}$ is the maximum number of iterations. $X_{i,k}$ is the position information of the $i$th sparrow in the $k$th dimension. $\alpha$ is a random number $(0, 1)$. $R_2$ is the warning value $(0, 1)$; $ST$ is the safe value $(0.5, 1]$. $Q$ is a random number. $L$ represents a $1 \times m$ matrix.

From Equation (12), when $R_2 < ST$, it means that the discoverer has not found that there are predators around the current foraging environment. At this time, the search space is safe and the discoverer can continue to perform more extensive search. When $R_2 \geq ST$, it means that there are predators. The discoverer will quickly send an alarm and send a signal to other sparrows. At this time, all sparrows will fly to other safe places to find food.

### 2.3.2. Update Follower Position

When foraging, the behavior of the discoverer will be watched by some followers. If the former finds better food, the latter will quickly detect it and immediately go to fight for food. The location update is described as follows:

$$X_{i,k}^{t+1} = \begin{cases} Q \cdot \exp\left(\frac{X_w - X_{i,k}^t}{i^2}\right), & i > n/2 \\ X_P^{t+1} + \left| X_{i,k} - X_P^{t+1} \right| \cdot A^+ \cdot L \end{cases} \tag{13}$$

where $X_P$, $X_w$ is the current best and worst position of the discoverer. $A$ is a $1 \times m$ matrix, and $A^+ = A^T \left(AA^T\right)^{-1}$. When $i > n/2$, it means that the $i$th follower has not found food, and it needs to continue to look for food.

### 2.3.3. Update the Guard Position

For the convenience of expression, we call these sparrows who are in danger without food as vigilantes. In the simulation, the number of vigilantes accounts for 10–20% of the total. The location update is described as follows:

$$X_{i,k}^{t+1} = \begin{cases} X_b^t + \beta \cdot \left| X_{i,k}^t - X_b^t \right|, & f_i \neq f_b \\ X_{i,k}^t + K \cdot \left( \dfrac{\left| X_{i,j}^t - X_w^t \right|}{(f_i - f_w) + \varepsilon} \right), & f_i = f_b \end{cases} \tag{14}$$

where $X_b$ is the current global optimal position. $\beta$ and $K$ are step control parameters. $f_i$ represents the fitness value of the current sparrow individual. $f_b$, and $f_w$ represent the current global optimal and worst fitness values, respectively. $\varepsilon$ is the minimum constant.

From Equation (15), if $f_i \neq f_b$, it indicates that the vigilant is at the edge of the population and is easily attacked by predators. If $f_i = f_b$, it indicates that the watcher is at the center of the population, and this part of sparrows has realized the threat. To prevent the predator from attacking, it must be close to other sparrows to reduce the risk of being attacked.

According to the design of sparrow search algorithm, the parameter optimization process of SSA is shown in Figure 8.

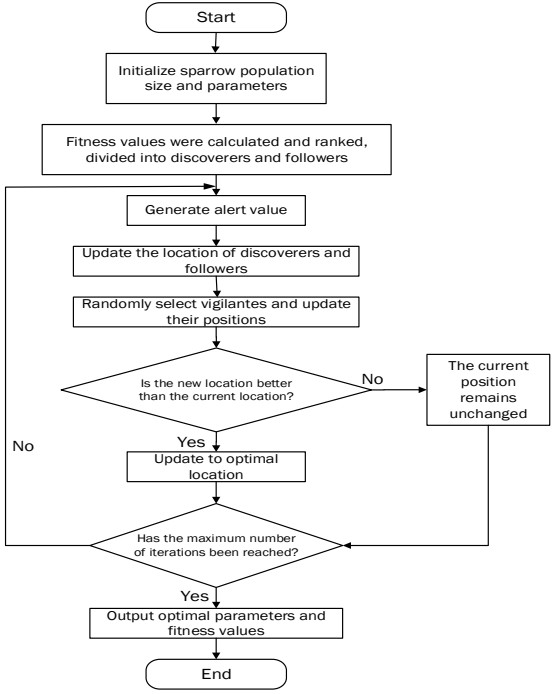

**Figure 8.** Flow chart of sparrow search algorithm.

### 2.4. Comparison of Optimization Algorithms

In order to test the optimization ability of sparrow search algorithm, particle swarm optimization (PSO), genetic algorithm (GA), and artificial bee colony algorithm (ABC) are introduced for experimental test and comparison. We test and compare the fitness values of the four algorithms through the Griewank multi peak test function. The dimension of the Griewank test function is set to 30 and the search range is $[-600, 600]$. The maximum number of iterations of each algorithm is set to 1000 and the population number is set to 100. The parameter settings of the four optimization algorithms are shown in Table 2:

**Table 2.** Optimization algorithm parameter setting.

| Optimization Algorithm | Set Value |
| :---: | :---: |
| GA | Crossing probability = 0.8; Variation probability = 0.05 |
| PSO | Acceleration factor c1, c2 = 1.5; Inertia factor w = 0.8 |
| ABC | Maximum mining times of honey source = 100 |
| SSA | The discoverers account for 20%, the vigilantes account for 10%, ST = 0.6 |

After the test function is tested, the fitness value of each algorithm is shown in Figure 9:

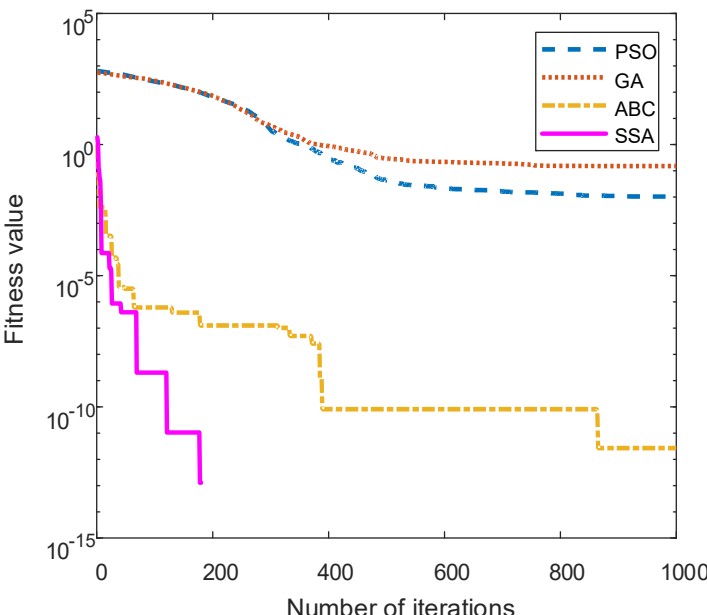

**Figure 9.** Fitness value of each optimization algorithm.

In the comparison of the four algorithms, the SSA algorithm has the fastest convergence speed and the highest convergence accuracy. The SSA algorithm obtains the best fitness value at the fastest speed in the iterative process, and its optimization ability is the best. It can be concluded that it has the advantages of high search accuracy, fast convergence speed, and strong stability. Therefore, the SSA algorithm is used to optimize the neural network parameters in this paper.

*2.5. CEEMD*

Empirical mode decomposition (EMD) is an adaptive data mining method for signal analysis. It analyzes the signal based on the time scale characteristics of the data themselves, and decomposes the original signal into a series of intrinsic mode components (IMF) and a residual component. However, the EMD method has serious mode aliasing. In 2010, Yeh et al. proposed the complementary set empirical mode decomposition algorithm (CEEMD), which is an improved algorithm of EMD and can solve this phenomenon.

CEEMD changes the extreme point of the original signal by adding a pair of white noise signals with opposite signs, and cancels the noise in the signal through multiple average processing. The decomposition process is shown in Figure 10.

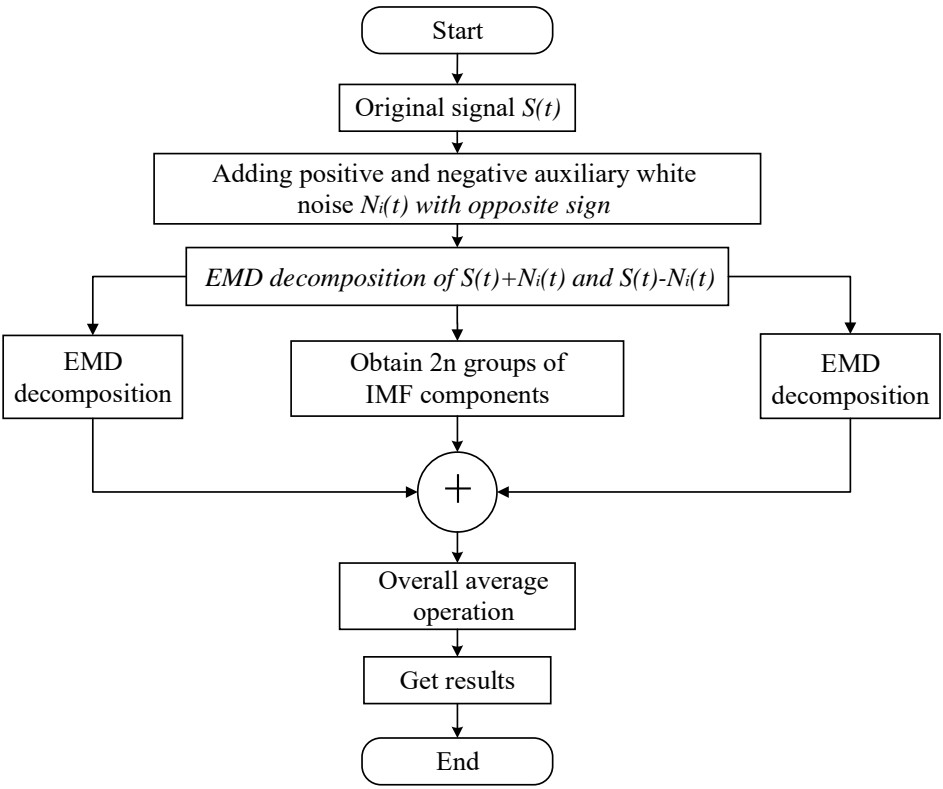

**Figure 10.** CEEMD decomposition flow chart.

(1)  First, *n* groups of white noise with opposite signs are added to the original signal $S(t)$ to obtain a pair of new signals, which can be expressed as shown in Equation (15):

$$\begin{bmatrix} M_{i1}(t) \\ M_{i2}(t) \end{bmatrix} = \begin{bmatrix} 1 & 1 \\ 1 & -1 \end{bmatrix} \begin{bmatrix} S(t) \\ N_i(t) \end{bmatrix} \tag{15}$$

where $N_i(t)$ represents added white noise; $M_{i1}(t)$, $M_{i2}(t)$ denotes signals obtained by adding positive and negative white noise, respectively;

(2)  Then, EMD decomposition is performed on the 2n signals obtained, and a group of IMF components are obtained for each signal, and the *j*th IMF component of the *i*th signal is recorded as $C_{ij}$; the last IMF component is taken as the residual component RES;

(3)  Finally, the 2*n* groups of IMF components obtained are averaged, and the components obtained by CEEMD decomposition of the original signal $S(t)$ are expressed as:

$$IMF_j = \frac{1}{2n} \sum_{i=1}^{2n} C_{ij} \tag{16}$$

where $IMF_j$ represents the *j*th IMF component obtained after decomposition.

## 3. Combined Forecasting Model

### 3.1. Introduction to Combination Model

The essence of CEEMD–SSA–GRU model prediction is equivalent to adding the complementary set empirical mode decomposition algorithm CEEMD on the basis of the SSA–GRU prediction model. From the original training and prediction of the training set load data directly, the CEEMD algorithm decomposes the training set load data to obtain several subsequences, and then predicts through the SSA–GRU prediction model.

### 3.2. Model Example Analysis

In the process of CEEMD decomposition, the signal-to-noise ratio Nstd of 0.01–0.5, the number of white noise additions NR of 50–300, and the parameter value of the maximum iteration number Maxiter of no more than 5000 are usually added to obtain a good decomposition effect. After several decomposition tests, the parameter values selected for the final decomposition in this paper are set as Nstd = 0.2, NR = 200, and Maxiter = 5000. This section selects the preprocessed training set load data in Section 2.2.2, and decomposes the training set data with CEEMD and EMD algorithms. The decomposition results are shown in Figures 11 and 12.

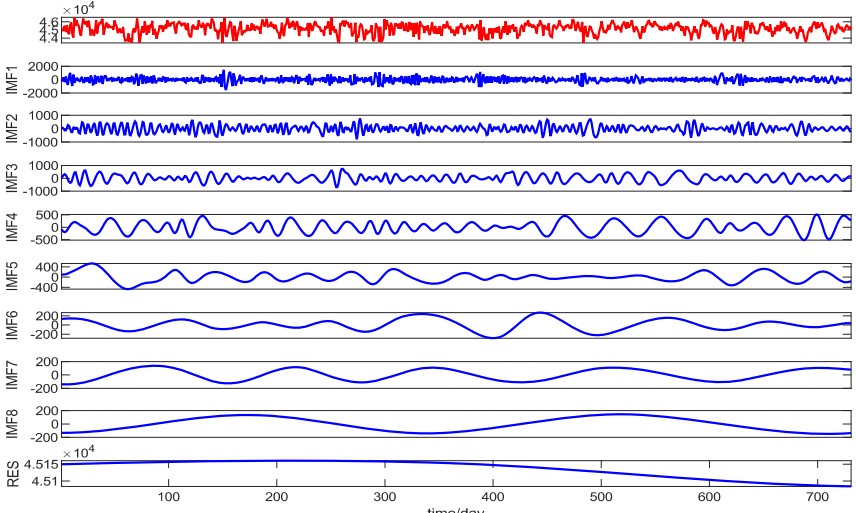

**Figure 11.** The modal components are obtained by EMD decomposition.

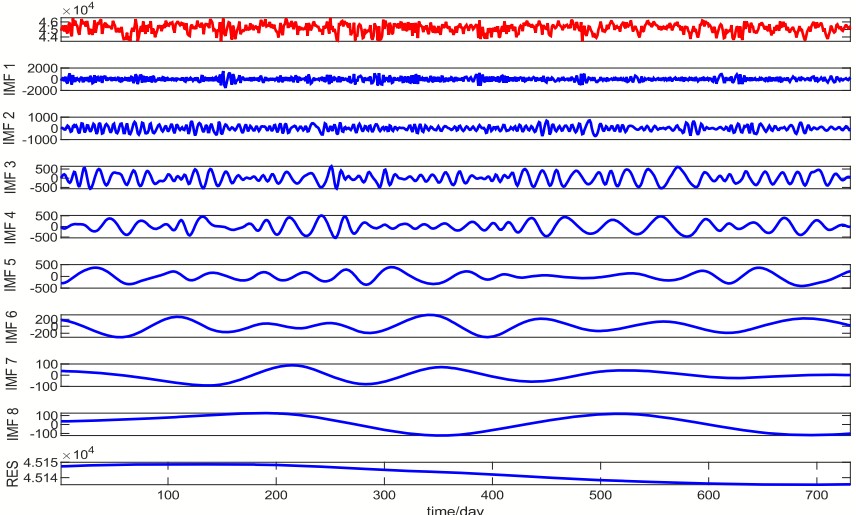

**Figure 12.** CEEMD decomposition to obtain modal components.

The prediction results based on CEEMD prediction model and EMD prediction model are compared and analyzed.

The parameter settings of the CEEMD algorithm are the same as those above. The range of parameters $n, \varepsilon, v, \beta$ in the SSA–GRU model are initialized to [10, 200], [0.001, 0.01], [50, 256], and [100, 1000]. The SSA initialization parameters are set according to Table 3. Through the construction of CEEMD–SSA–GRU model and EMD–SSA–GRU model, the experimental simulation is carried out on the two models to predict the data in the next

month. The prediction results of the two models are shown in Table 4, and the result curves are shown in Figures 13 and 14.

**Table 3.** SSA initialization parameter setting.

| Relevant Parameters | Set Value |
|---|---|
| Population number | 20 |
| Number of iterations | 50 |
| Safety threshold | 0.6 |
| Number of discoverers | 20% |
| Number of vigilantes | 10% |

**Table 4.** Comparison of prediction results of two models.

| Time/ Day | Actual Value (kw·h) | EMD–SSA– GRU (kw·h) | CEEMD–SSA– GRU (kw·h) | Time/ Day | Actual Value (kw·h) | EMD–SSA– GRU (kw·h) | CEEMD–SSA– GRU (kw·h) |
|---|---|---|---|---|---|---|---|
| 1 | 45,360.37 | 45,753.35 | 45,392.99 | 17 | 45,424.44 | 45,613.16 | 45,806.91 |
| 2 | 46,338.27 | 46,518.48 | 46,188.03 | 18 | 46,514.46 | 45,677.67 | 46,150.98 |
| 3 | 45,877.85 | 46,151.58 | 45,953.33 | 19 | 45,238.45 | 45,553.17 | 45,332.02 |
| 4 | 45,942.52 | 45,693.28 | 45,853.42 | 20 | 45,856.65 | 45,603.49 | 45,426.96 |
| 5 | 45,840.91 | 46,234.02 | 46,114.57 | 21 | 45,862.76 | 45,814.70 | 45,545.24 |
| 6 | 46,291.17 | 46,362.10 | 46,330.02 | 22 | 45,593.45 | 45,775.15 | 45,730.47 |
| 7 | 45,471.49 | 45,632.27 | 45,620.18 | 23 | 45,831.85 | 45,859.43 | 45,682.67 |
| 8 | 45,142.91 | 44,709.40 | 45,187.37 | 24 | 45,343.99 | 46,151.51 | 45,902.67 |
| 9 | 44,379.54 | 44,644.54 | 44,911.17 | 25 | 44,872.57 | 45,458.34 | 45,604.36 |
| 10 | 45,039.70 | 44,519.86 | 44,767.94 | 26 | 46,133.70 | 46,080.31 | 46,012.50 |
| 11 | 44,262.69 | 45,314.84 | 45,137.64 | 27 | 44,743.78 | 45,284.60 | 45,123.53 |
| 12 | 45,438.92 | 45,243.34 | 45,096.62 | 28 | 45,927.41 | 45,341.90 | 45,429.72 |
| 13 | 45,054.37 | 45,313.75 | 45,616.59 | 29 | 45,654.40 | 45,394.16 | 45,626.17 |
| 14 | 44,940.83 | 45,764.07 | 45,349.38 | 30 | 45,462.24 | 45,661.05 | 45,312.03 |
| 15 | 45,883.15 | 45,625.53 | 45,677.65 | 31 | 45,460.24 | 46,165.21 | 45,331.24 |
| 16 | 44,214.98 | 44,083.94 | 44,675.42 | | | | |

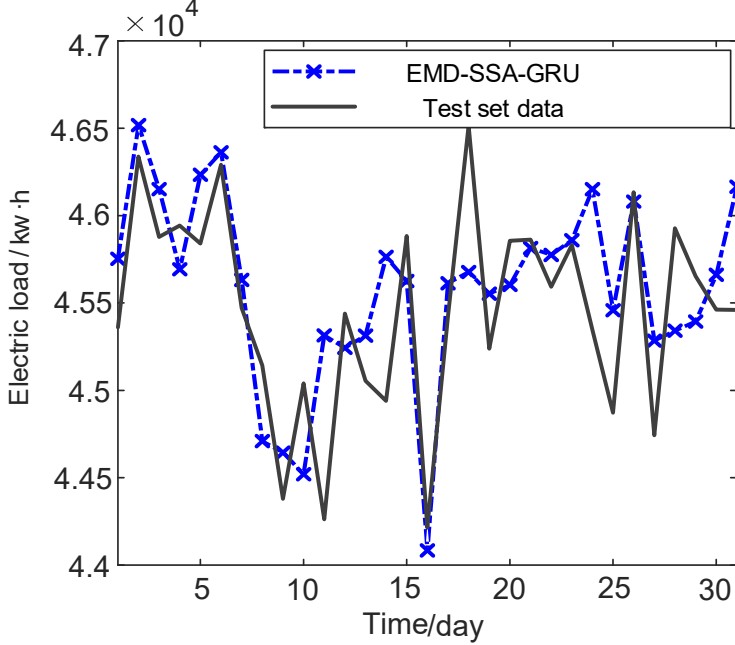

**Figure 13.** EMD–SSA–GRU model prediction data.

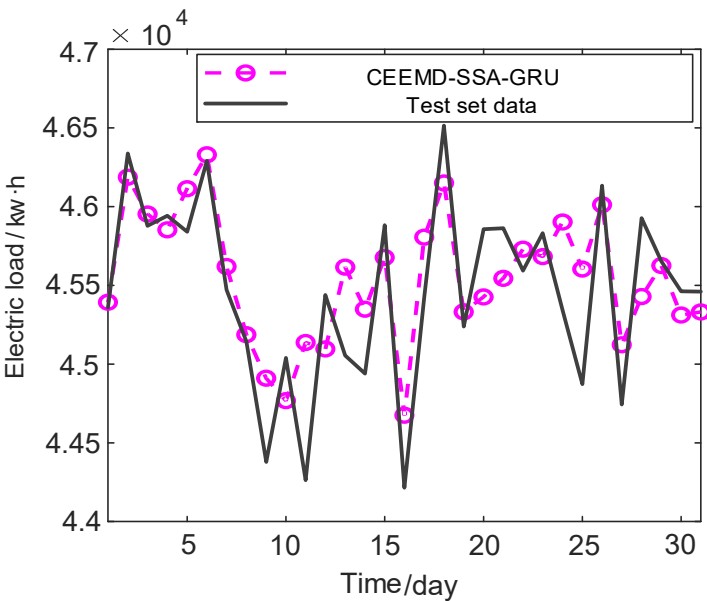

**Figure 14.** CEEMD–SSA–GRU model prediction data.

The curve fitting degree of the CEEMD–SSA–GRU model is higher than that of the EMD–SSA–GRU model, and the number of extreme points close to the real value is higher. In order to more intuitively see the prediction of the two models, the prediction errors of the two models are calculated. The comparison of the errors of the two models is shown in Figure 15.

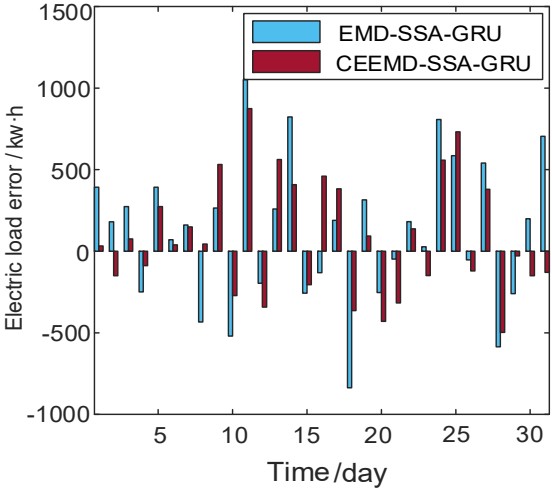

**Figure 15.** Comparison of power load error.

For the maximum relative error and average absolute error, the EMD–SSA–GRU model is 2.038% and 0.80%, respectively; the CEEMD–SSA–GRU model is 1.98% and 0.64%, respectively. Through comparison, the accuracy of the maximum relative error of CEEMD–SSA–GRU model is increased by 16.8%, and the accuracy of the average relative error is increased by 20.0%. From the most direct prediction error analysis, we can see that the CEEMD–SSA–GRU model has higher prediction accuracy and accuracy.

Each error evaluation index formula calculates each error of the prediction results of the two models, as shown in Table 5.

**Table 5.** Prediction error evaluation index.

| Model | MAPE | MAE (%) | RMSE (%) |
|---|---|---|---|
| EMD–SSA–GRU | 0.0080 | 3.63 | 4.47 |
| CEEMD–SSA–GRU | 0.0064 | 2.90 | 3.60 |

Compared with the EMD–SSA–GRU model, the accuracy of the MAPE, MAE, and RMSE of the CEEMD–SSA–GRU prediction model increases by 20.0%, 20.1%, and 19.5%, respectively.

## 4. Results

We compare the prediction effect of CEEMD–SSA–GRU model with that of single GRU model, the GRU model optimized by SSA, and the EMD–SSA–GRU model. The comparison of the prediction curves of the four models is shown in Figure 16 and the comparison of prediction errors is shown in Figure 17.

According to the calculation of error evaluation index formula, the comparison of various error evaluation indexes is shown in Table 6.

**Table 6.** Prediction error evaluation index.

| Model | MAPE | MAE (%) | RMSE (%) |
|---|---|---|---|
| GRU | 0.0163 | 7.40 | 8.82 |
| SSA–GRU | 0.0105 | 4.76 | 5.86 |
| EMD–SSA–GRU | 0.0080 | 3.63 | 4.47 |
| CEEMD–SSA–GRU | 0.0064 | 2.90 | 3.60 |

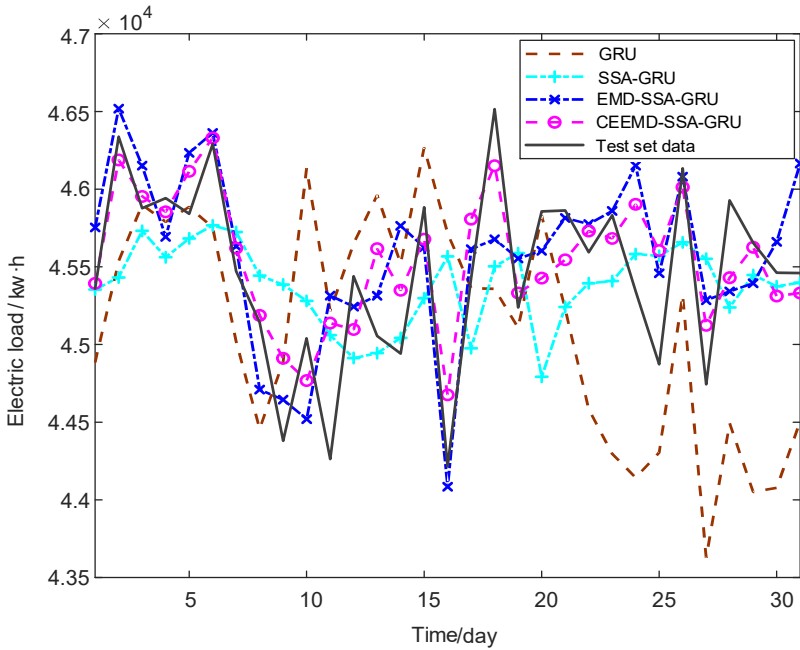

**Figure 16.** Comparison of prediction curves of four models.

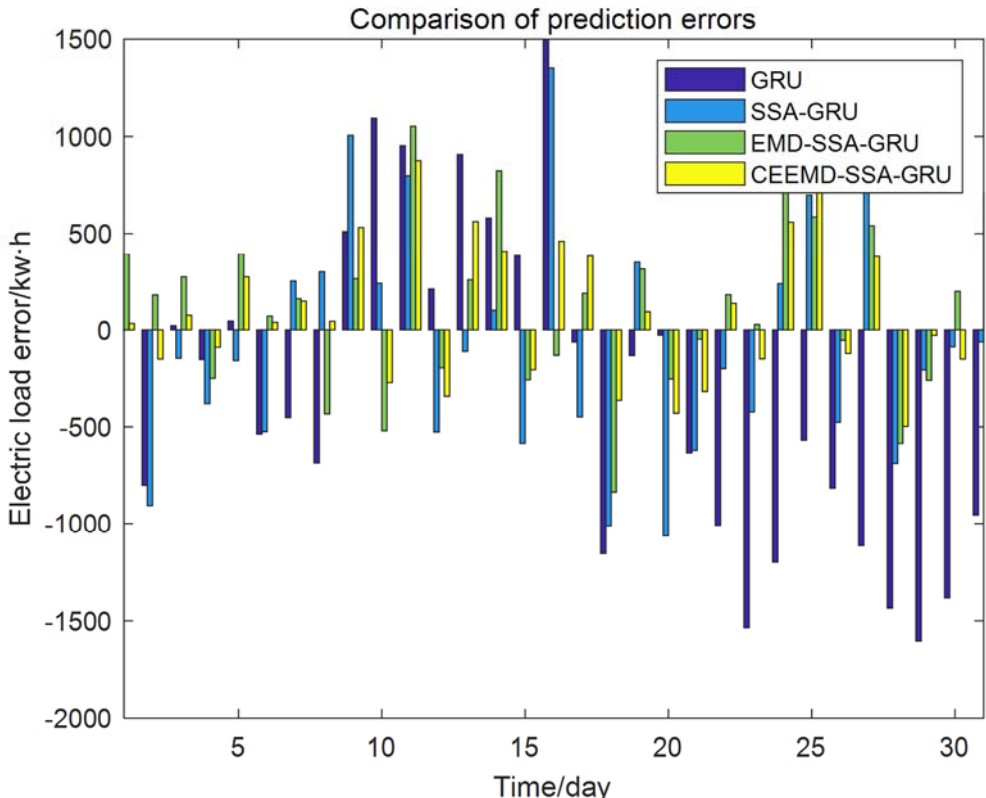

**Figure 17.** Comparison of prediction errors of four models.

## 5. Conclusions

The fitting degree of the prediction curve of each model from high to low is the CEEMD–SSA–GRU model, the EMD–SSA–GRU model, the SSA–GRU model, and the GRU model, and the predicted value of each point in the prediction curve of CEEMD–SSA–GRU model is closest to the extreme point of the real curve. It can be seen from Table 4 that for MAPE, the CEEMD–SSA–GRU model is 60.7% lower than the GRU model, 39.0% lower than the SSA–GRU model, and 20.0% lower than the EMD–SSA–GRU model. For MAE, the CEEMD–SSA–GRU model is 60.8% lower than the GRU model, 39.1% lower than the SSA–GRU model, and 20.1% lower than the EMD–SSA–GRU model. For RMSE, the CEEMD–SSA–GRU model is 59.2% lower than the GRU model, 38.5% lower than the SSA–GRU model, and 19.5% lower than the EMD–SSA–GRU model.

The prediction accuracy of the CEEMD–SSA–GRU model reaches 99.36%, and the prediction result of the CEEMD–SSA–GRU model is the most accurate. Its prediction accuracy is obviously better than the other three models, and the fitting degree of the curve is the closest to the real curve. Therefore, the CEEMD–SSA–GRU model has more advantages in short-term power load forecasting and can better provide reliable forecasting trends for industrial users.

**Author Contributions:** Conceptualization, C.L.; formal analysis, L.S.; methodology, C.L. and J.L.; resources, L.S.; supervision, J.L. and H.W.; validation, Q.G.; visualization, Q.G.; writing—original draft, Q.G. All authors have read and agreed to the published version of the manuscript.

**Funding:** This research received no external funding.

**Data Availability Statement:** The load forecasting data used to support the results of this study have not been provided because they are private data of enterprises.

**Conflicts of Interest:** The authors declare that there are no conflicts of interest regarding the publication of this paper.

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
