# Peer review of "Research on Short-Term Load Forecasting Based on Optimized GRU Neural Network"

_electronics, doi:10.3390/electronics11223834_

Round 1

Reviewer 1 Report

The manuscript “Research on short term load forecasting based on optimized GRU neural network” proposes a gated cyclic network model (SSA-GRU) based on sparrow algorithm optimization and compares the combined model CEEMD-SSA-GRU with EMD-SSA-GRU, SSA-GRU and GRU models. I think the manuscript needs some reworking, and the problems indicated below must be addressed:

1. Literature analysis should be deepened

2. The science gap should be described in more detail, and the purpose of the work should also be emphasized more.

3. Are the drawings an own study?

4. Chapter 2.2.1 are three formulas, please indicate why these criteria have been chosen and describe them in detail.

5. There is no need to insert figures 6 and 7, because the same data are presented in figure 8.

6. In Table 1. Comparison of BP and GRU neural network prediction results. - the results of the operation of two models are given, however, there are no learning characteristics of these models, learning parameters, e.g. how the data were divided, etc., and the results, e.g. MAE, RMSE, MAPE, and the correlation coefficient.

7. What software was used?

8.  There is no discussion in the article, please add it.

9. Conclusions should be expanded.

Author Response

  1. See the attachment to the revised article
  2. See the attachment to the revised article
  3. Yes, it's all my own research and lab results.
  4. The mean absolute error (MAE) is the average of the absolute sum of the difference between the predicted value and the true value. MAE values range from 0 to infinity. The smaller the value, the better the prediction effect.
    The root mean square error (RMSE) adds a square root to the mean square error (MSE). MSE is the ratio of the square sum of the deviation between the true value and the predicted value to the number of predictions. The value of RMSE ranges from 0 to infinity. The smaller the value is, the better the prediction effect of the prediction model is. The larger the value, the greater the error.
    Mean Absolute percentage error (MAPE) is the product of mean relative error and percentage and is often used to describe how discrete the data is. The value of MAPE ranges from 0 to infinity. A value of MAPE of 0 indicates that the prediction model is perfect. If the value of MAPE is greater than 1, it is a poor model.
  5. See the attachment to the revised article
  6. The comparison results of MAE, RMSE and MAPE of the two models were added.
  7.  Matlab2021a

Reviewer 2 Report

The authors do not clearly present their contribution. It seems that the paper is an application of 3 combined CEEMD-SSA-GRU techniques, each already existing and developed for similar purposes.
It is disturbing to see prediction signals totally disagreeing with the real signal (figs 6 and 7) but with errors of the order of a few percent. In fact, the initial signal shows very little relative variation, [43000, 47000] kW.h since the signal is normalized between 0 and 1, a modeling error of 100% will only lead to a relative error between 43000 and 47000, (47000-43000)/40000 = 10% maximum
Then, authors should test their prediction model with more classical power consumption signals, showing week-end variation for instance. It would also be important to show that the approach works with hourly time steps rather than daily energies.

Author Response

The data of BP model in Figure 6 has not obvious prediction law and effect, which is quite different from the actual data. It is for this reason that all the authors have come up with neural network models that make better predictions. The prediction curve of the GRU model in Figure 7 is roughly consistent with the trend of the actual data, but some data points are quite different, so the author further optimized to achieve better prediction effect.

As for the questions you raised, the author is also aware of these problems and will study and improve some of them in the future. In the final discussion and conclusion, the author will raise relevant issues and improve them in the future research. Thank you for your question. I'll pay attention.

Although the prediction model proposed in this paper improves the prediction accuracy to a certain extent, the work in this paper still has some shortcomings, which can be studied and improved from the following two aspects in the future:

(1) This paper only considers the characteristics of load data itself. However, some electrical loads are affected by certain factors that need to be taken into account when forecasting these load data, such as weekend factors. Therefore, it is necessary to predict different load data through the model to verify the universality of the model.

(2) The model in this paper has achieved good results in short-term power load prediction, but there is still room for research in the medium and long term.

Author Response

1.Please refer to the attachment for the corrected article

2.Please refer to the attachment for the corrected article

3.(1) Data completion
Power load data may be missing due to some special reasons or circumstances, so it is necessary to complete the original data. The method is to take the average value of the load data at the same time of the two adjacent days with the missing value.
(2) Outlier processing
Due to the influence of equipment, data transmission environment and other factors, there may be outliers in the original data, which will affect the accuracy of the prediction. Therefore, such cases should be handled. The method is to delete outliers and perform mean replacement.
(3) Data normalization
In the process of training the training set data, the neural network model will be sensitive to the input training set data scale, and the high value will affect the training effect of the model. In this case, the normalized method can be used to process the completed data.

4.Modify according to expert advice.

5.Modify according to expert advice.

6.Modify according to expert advice.

A comparison of relative errors can be added, and the maximum and minimum, average relative errors can be compared.